# Study on Lavender Essential Oil Chemical Compositions by GC-MS and Improved pGC

**DOI:** 10.3390/molecules25143166

**Published:** 2020-07-10

**Authors:** Guangyao Dong, Xiaohui Bai, Aoken Aimila, Haji Akber Aisa, Maitinuer Maiwulanjiang

**Affiliations:** 1The Key Laboratory of Plant Resources and Chemistry of Arid Zone, Xinjiang Technical Institute of Physics and Chemistry, Chinese Academy of Sciences, Urumqi 830011, China; donggh17@ms.xjb.ac.cn (G.D.); baixiaohui18@mails.ucas.ac.cn (X.B.); aimila16@ms.xjb.ac.cn (A.A.); haji@ms.xjb.ac.cn (H.A.A.); 2School of Chemical Science, University of the Chinese Academy of Sciences, Beijing 100039, China; 3State Key Laboratory Basis of Xinjiang Indigenous Medicinal Plants Resource Utilization, Xinjiang Technical Institute of Physics and Chemistry, Chinese Academy of Sciences, Urumqi 830011, China

**Keywords:** lavender, essential oil, improved preparative GC, NMR

## Abstract

Lavender essential oil from the aerial parts of *Lavandula angustifolia* Mill. was analyzed by GC-MS equipped with three capillary columns of different polarities, which were HP-1, HP-5 ms and HP-INNOWax. A total of 40 compounds were identified by GC-MS, accounting for 92.03% of the total essential oil compositions. Nineteen monomers were separated by column chromatography and improved preparative gas chromatography (pGC), six of which could not be retrieved from the NIST 14 (National Institute of Standards and Technology, USA; 14th edition) library database. Fifteen compounds were identified for the first time in lavender essential oil. The improved pGC not only doubled the efficiency but also greatly reduced the cost.

## 1. Introduction

Lavender (*Lavandula angustifolia* Mill.) is a perennial herb and a member of the Lamiaceae (Labiatae) family that is native to the Mediterranean region and has been grown all over the world because of the huge market for essential oils. Lavender essential oil is widely used in fragrances and commodities including colognes, skin lotions, soaps, food flavorings, perfumes and aromatherapeutic medicines [1]. Lavender is also widely cultivated in China, and its main producing area is in Yili, Xinjiang. The *Lavandula* genus is divided into 37 varieties according to the shape of the leaves, corolla morphology, calyces and bract in “Lavender: The genus Lavandula” [2]. Only the essential oils of three lavender species (*Lavandula angustifolia*, *Lavandula latifolia* and *Lavandula hybrid*) play an important role in the perfume and cosmetics industry [3]. The essential oils of lavender are mainly produced from glands on the surface of the flowers and the leaves [2].

Although the chemical compositions of lavender essential oil were studied by GC-MS in different literatures [4,5], there is no report on the separation and analysis of lavender essential oil by preparative gas chromatography (pGC), NMR and GC-MS equipped with three capillary columns of different polarities. This study could not only verify experimental results, but also the isolated monomer by pGC is of great help to the study of the activity of the single and composite component of lavender essential oil. The combination of these two methods can effectively solve the problem of identifying the accuracy of essential oil components by GC-MS without reference compounds. Therefore, the study has important analytical significance for identifying essential oil components.

Many reports published have shown lavender essential oil to possess high cholinesterase inhibitory activities [6,7] and other biological activities beneficial to human health, such as being antibacterial, antifungal [8], sedative [9], anti-depressive, effective for burns and insect bites [10], anticancer [11], anti-spasmolytic, anti-inflammatory [12], antioxidant [13], acaricides etc. [14]. The main component of essential oils are terpenoids, which are low-molecular and lipophilic compounds that can easily cross the blood-brain barrier [15]. The compounds separated from lavender essential oil could lay the foundation for subsequent drug research.

## 2. Results and Discussion

### 2.1. Chemical Composition of the Essential Oils

The essential oil was extracted by hydrodistillation from the fresh plants of lavender aerial parts, and the mean values of the oil yields were 1.3% (*v*/*w*; mL/g) based on fresh weights. The essential oil GC-MS results of three different polarity capillary columns, which were HP-5 ms, HP-1 and HP-INNOWax capillary columns, are presented in Table 1 and Appendix A. The relative content of the essential oil components was normalized by peak area and expressed as a percentage. The integration method is that 0.8% of the maximum peak area was selected as the minimum integral peak. According to the above integration method, 40 peaks could be identified by comparing their real retention indices relative to *n*-alkanes (C_7_–C_30_) and mass spectra with the NIST 14 (National Institute of Standards and Technology, USA; 14th edition) Mass Spectral Library, which accounts for 92.03% of the total amount of essential oil compounds. The ninth peak can be determined as the mixed peak of the *cis* and *trans* isomer of the linalool oxide according to the GC-MS results of the HP-1 and HP-INNOWax capillary columns.

Forty compounds were identified in the essential oil by GC-MS, representing 92.03% of the total volatiles (Table 1). The identified monomers belong to different chemical classes and are present in different proportions. The essential oil contains essentially oxygenated monoterpenes (31.53%) and esters (43.23%), significant fraction monoterpene hydrocarbons (8.03%), sesquiterpene hydrocarbons (3.61%), oxygenated sesquiterpenes (4.54%), and small quantities of other compounds (1.14%). Linalool (**12**, 19.71%)—an oxygenated monoterpene—and linalyl acetate (**27**, 26.61%) and lavandulol acetate (**32**, 12.68%)—ester compounds—were the main components of the lavender essential oil.

The compounds that were separated by silica gel column chromatography and pGC were analyzed and identified by NMR (Appendix A. Twelve compositions, which were compounds **9**, **10**, **12**, **13**, **17**, **18**, **21**, **27**, **32**, **38**, **39**, **40** and **43**, were further identified by comparing the carbon spectrum data in the literature [16,17,18,19,20,21,22,23,24,25,26,27,28]. Compared with the method of determining compounds by GC-MS, six missing compositions, which were compounds **19**, **29**, **33**, **41**, **45** and **46**, were identified by comparing the carbon spectrum data in the literature [29,30,31,32,33,34], and could not be retrieved from the NIST 14 database. By comparison with the literature [35], 15 compounds were newly identified. In this study, sulfur was isolated from essential oil for the first time, and was identified by GC-MS, whose mass spectrogram is shown in Appendix A. Therefore, it was proved that lavender essential oil contained a small amount of sulfur compounds. The sulfur compounds provide a new explanation for the insecticidal activity of lavender essential oil and provide a new research idea for later researchers. This finding may help to develop lavender essential oil into a new natural acaricides [36].

### 2.2. Improvements in the Fraction Collector of pGC

In the early stages of the separation of lavender essential oils by pGC, we found that the cost and separation efficiency of this instrument were too low. To solve this problem, we converted the liquid nitrogen cooling system of the fraction collector to a cold trap cooling system. This improvement not only increased efficiency but also greatly reduced costs.

By improving the preparation fraction collector of pGC without affecting the normal operation of the pGC, it was found that the improved pGC had a similar collection amount in the same time and temperature compared with the unmodified equipment. The liquid nitrogen cooling system of the preparative fraction collector of pGC was transformed into a cold well cooling system. This improvement doubled the efficiency of preparative gas chromatography, and the cost of the cooling system was reduced from RMB 675/day to RMB 25/day.

## 3. Materials and Methods

### 3.1. Plant Material and Reagents

The aerial parts of *Lavandula angustifolia* Mill. were collected in June 2019 in Yili, Xinjiang, China. The plant material was dried in the shade. The plant sample was identified by Dr. Chunfang Lu and the voucher specimen was stored in the Xinjiang Technical Institute of Physics and Chemistry, Urumqi, Chinese Academy of Sciences (voucher species man No. WY02260). Anhydrous sodium sulfate was purchased from Tianjin Hongyan chemical reagent factory (Tianjin, China), *n*-hexane, ethyl acetate, and acetic acid were purchased from Tianjin Yongsheng Superfine Chemical Industry Co., Ltd. (Tianjin, China). *n*-Alkanes (C_7_–C_30_) were purchased from Sigma-Aldrich (Shanghai, China).

### 3.2. Extraction of the Essential Oil

The lavender essential oil was obtained from aerial parts of the lavender that were cut into about 2 cm by hydrodistillation for 3 h using a self-assembling Clevenger-type apparatus, and the ratio of material to liquid was 1:10. The essential oil was collected, centrifuged, and taken from the upper layer. The lavender essential oil was extracted three times with 1200 g of material each time. The yields were 12.9852 g, 13.9137 g and 11.5129 g, respectively. The upper layer (essential oil) was dried over Na_2_SO_4_ and stored at −20 °C until required for analysis and separation.

### 3.3. Gas Chromatography Analysis

The analysis of the essential oil was first performed on an Agilent GC-QTOF-MS system consisting of a 7890B gas chromatograph equipped with HP-5 ms capillary column (30 m × 0.25 mm i.d., film thickness 0.25 µm) and 7693 autosampler, connected to a hybrid QTOF mass spectrometer (Agilent model 7200, Santa Clara, CA, USA), controlled by MassHunter Acquisition B.07.00 software (Santa Clara, CA, USA).

Helium was used as a carrier gas at a flow rate of 1 mL/min. The injector and detector temperatures were 250 °C. The oven temperature was programmed from 70 (10 min) to 100 °C at a rate of 2 °C/min, then from 100 to 200 °C at a rate of 5 °C/min. The injection volumes were 0.4 µL. The split injection was conducted with a split ratio of 1:40. The mass spectra were recorded at 70 eV (EI) and were scanned in the range 50–500 *m*/*z*.

The essential oil was analyzed on the Agilent GC-MS system consisting of a 7693 autosampler and 7890A gas chromatograph connected to a 5975C mass spectrometer (Santa Clara, CA, USA) (inert XL EI/CI MSD with Triple-Axis detector), controlled by 5975-7890GC-MS software (Santa Clara, CA, USA). The GC was equipped with HP-INNOWax (30 m × 0.25 mm i.d., film thickness 0.25 µm) and HP-1 capillary columns (30 m × 0.25 mm i.d., film thickness 0.25 µm). Nitrogen was used as a carrier gas at a flow rate of 0.8 mL/min. The injector and detector temperatures were 250 °C. The oven temperature was programmed from 60 to 180 °C at a rate of 8 °C/min, then from 180 to 240 °C (5 min) at a rate of 10 °C/min. The injection volumes were 0.8 µL. The split injection was conducted with a split ratio of 1:40. The mass spectra were recorded at 70 eV (EI) and were scanned in the range 30–500 *m*/*z*. The components were identified by comparing their real retention indices relative to the *n*-alkanes (C_7_–C_30_) and the mass spectra with the NIST 14 Mass Spectral Library. The formula for calculating the retention index is as follows [36]:(1)$$\mathrm{RI}\;=100Z\;+100\;\left(T_{x}-{\;T}_{z}\right)/\left(T_{z+1}-{\;T}_{z}\right)$$
where T_X_ is the component retention time; Z is the carbon number of *n*-alkanes component; T_Z_ is the the retention time of *n*-alkanes that carbon number is Z; and T_Z+1_ is the the retention time of *n*-alkanes that carbon number is Z + 1.

### 3.4. Isolation and Structure Elucidation

The pGC was modified based on the Agilent 7890B gas chromatograph system (Agilent model 7200, Santa Clara, CA, USA). It was equipped with a HP-5 capillary column (30 m × 0.53 mm i.d., film thickness 1.0 µm), a G4513A autosampler, a flame ionization detector (FID), a modular analytical system and a preparative fraction collector (Gerstel Company, Mülheim, Germany) equipped with a home-made cold trap cooling system.

In brief, 30.0 g of essential oil was firstly separated by silica gel column chromatography. The separation conditions were 380 g, 200–300 mesh silica gel and 60 × 600 mm column volume. The elution gradients were *n*-hexane:ethyl acetate = 100:0, 100:1, 100:1.5, 100:3, 100:8, 100:15, 100:30, 100:50, 0:100. According to the results of the thin layer, divided appropriately, the fractions were condensed to 4 mL and stored in brown vials at −20 °C. Then, suitable fractions were selected to separate monomers by improved pGC. Seven fractions were selected to separate the compounds by pGC, which were successively named A-G. Compound 38 (3.0 mg), compound 39 (1.3 mg), and compound 40 (0.8 mg) were prepared from A fractions by pGC. The injection volumes were 0.5 µL. The number of injections was 466. The oven temperature of pGC was programmed from 70 to 160 °C at a rate of 50 °C/min, then from 160 to 174 °C at a rate of 2 °C/min, and then from 174 to 280 °C (0.5 min) at a rate of 50 °C/min. Compounds 13 (1.5 mg), 18 (1.9 mg), 21 (3.7 mg) and 41 (0.7 mg) were prepared from B fractions by pGC. The injection volumes were 0.5 µL. The number of injections was 411. The oven temperature of pGC was programmed from 70 to 150 °C at a rate of 10 °C/min, then from 150 to 250 °C (0.5 min) at a rate of 50 °C/min. Compounds 27 (25.7 mg), 32 (15.3 mg), 43 (3.5 mg) and 46 (1.4 mg) were prepared from C fractions by pGC. The injection volumes were 1.0 µL. The number of injections was 524. The oven temperature of pGC was programmed from 100 to 170 °C at a rate of 10 °C per min, then from 170 to 250 °C (1.5 min) at a rate of 30 °C/min. Compounds 18 (3.8 mg), 17 (4.2 mg) and 41 (0.3 mg) were prepared from D fractions by pGC. The injection volumes were 1.0 µL. The number of injections was 435. The oven temperature of pGC was programmed from 150 (5 min) to 230 °C at a rate of 20 °C/min (1.5 min). Compounds 10 (5.2 mg) and 33 (2.7 mg) were prepared from E fractions by pGC. The injection volumes were 1.0 µL. The number of injections was 376. The oven temperature of pGC was programmed from 140 to 150 °C at a rate of 2 °C/min, then from 150 to 180 °C at a rate of 10 °C/min, then from 180 to 280 °C (5 min) at a rate of 100 °C/min. Compound 29 (27.5 mg) was prepared from F fractions by pGC. The injection volumes were 5.0 µL. The number of injections was 241. The oven temperature of pGC was programmed from 100 to 240 °C (1.5 min) at a rate of 20 °C/min. Compound 45 (1.0 mg) was prepared from G fractions by pGC. The injection volumes were 4.0 µL. The number of injections was 431. The oven temperature of pGC was programmed from 100 to 200 °C at a rate of 25 °C/min, then from 200 to 250 °C (3 min) at a rate of 20 °C/min. The structure of the monomers was proved by ^13^C-NMR.

## 4. Conclusions

In this study, the chemical composition of the lavender essential oil obtained from the aerial parts was studied by pGC and GC-MS equipped with three capillary columns of different polarity for the first time. Using GC-MS equipped with three capillary columns to analyze lavender essential oil made the composition accuracy of lavender essential oil more valuable for reference. Compared with the method of determining compounds by GC-MS, six missing compositions, which were 2,2,6-trimethyl-6-vinyltetrahydro-2*H*-pyran-3-ol, 3,7-dimethylocta-1,7-diene-3,6-diol, (*E*)-7-hydroxy-3,7-dimethylocta-1,5-dien-ylacetate, 11-hydroxy-α-santal-9-ene, (3*S*,6*R*,9*R*)-2-(hydroxymethyl)-5,5,9-trimethyltricyclo[7.2.0.0(3,6)]undecan-2-ol and (6*R*,10*R*)-6,10,14-Trimethyl-2-pentadecanone, were identified. In comparison with the literature about lavender essential oil, 15 compounds were newly identified. The improved pGC not only doubled the efficiency but also greatly reduced the cost. This improved pGC also holds great promise.

## Figures and Tables

**Table 1 molecules-25-03166-t001:** Relative content and identified methods of the chemical compositions of lavender essential oil.

NO.		Compound Name	Molecular Formula	Lit. RI_HP-5_ ^a^	Exp. RI_HP-5 ms_ ^b^	Exp. RI_HP-1_	Exp. RI_HP-INNOWax_	Relative Content/% ^c^	Identified Methods
1		Camphene	C_10_H_16_	952	943	-	1088	0.41	GC-MS, RI
2		β-Myrcene	C_10_H_16_	991	989	982	1168	0.60	GC-MS, RI
3		*p*-Cymene	C_10_H_14_	1025	1035	1014	1285	0.23	GC-MS, RI
4		β-Cymene	C_10_H_14_	1023	1021	1011	1290	0.61	GC-MS, RI
5		Limonene	C_10_H_16_	1030	1024	-	1217	0.58	GC-MS, RI
6		Cineole	C_10_H_18_O	1032	1026	1021	1219	1.05	GC-MS, RI
7		β-*cis*-Ocimene	C_10_H_16_	1038	1033	1026	1240	3.31	GC-MS, RI
8		β-*trans*-Ocimene	C_10_H_16_	1049	1044	1038	1259	1.41	GC-MS, RI
9		*cis*-Linalool oxide	C_10_H_18_O_2_	1074	1069	1059	1455	0.49	GC-MS, RI, ^13^C-NMR
10		*trans*-Linalool oxide	C_10_H_18_O_2_	1086	1069	1074	1484		GC-MS, RI, ^13^C-NMR
11		α-Terpinolen	C_10_H_16_	1088	1086	-	1301	0.63	GC-MS, RI
12		Linalool	C_10_H_18_O	1099	1102	1090	1545	19.71	GC-MS, RI, ^13^C-NMR
13		Hotrienol	C_10_H_16_O	1107	1105	-	-	0.46	GC-MS, RI, ^13^C-NMR
14		1-Pentylallyl acetate	C_10_H_18_O_2_	1111	1114	1095	1380	1.01	GC-MS, RI
15		(4*E*,6*Z*)-allo-Ocimene	C_10_H_16_	1131	1128	-	-	0.25	GC-MS, RI
16		Camphor	C_10_H_16_O	1145	1140	1123	-	0.42	GC-MS, RI
17		(−)-Borneol	C_10_H_18_O	1167	1161	1152	1723	1.78	GC-MS, RI, ^13^C-NMR
18		Lavandulol	C_10_H_18_O	1170	1166	-	1681	0.48	GC-MS, RI, ^13^C-NMR
19		2,2,6-trimethyl-6-vinyltetrahydro-2*H*-pyran-3-ol	C_10_H_18_O_2_	-	1166	-	-	-	GC-MS, RI, ^13^C-NMR
20		4-Terpineol	C_10_H_18_O	1177	1173	1165	1621	0.41	GC-MS, RI
21		Cryptone	C_9_H_14_O	1184	1182	1160	1715	0.62	GC-MS, RI, ^13^C-NMR
22		α-Terpineol	C_10_H_18_O	1189	1187	1176	1712	3.61	GC-MS, RI
23		Verbenone	C_10_H_14_O	1205	1204	1185	-	0.22	GC-MS, RI
24		Bornyl formate	C_11_H_18_O_2_	1226	1225	-	-	0.41	GC-MS, RI
25		Neryl alcohol	C_10_H_18_O	1228	1229	1213	1804	0.49	GC-MS, RI
26		*p*-Cumic aldehyde	C_10_H_12_O	1239	1238	-	1823	0.72	GC-MS, RI
27		Linalyl acetate	C_12_H_20_O_2_	1257	1263	1242	1564	26.61	GC-MS, RI, ^13^C-NMR
28		Phellandral	C_10_H_16_O	1276	1273	1254	-	0.35	GC-MS, RI
29		3,7-dimethylocta-1,7-diene-3,6-diol	C_10_H_18_O	-	1276	-	-	-	GC-MS, RI, ^13^C-NMR
30		Bornyl acetate	C_12_H_20_O_2_	1285	1285	-	-	0.71	GC-MS, RI
31		Cuminol	C_10_H_14_O	1289	1290	-	-	0.33	GC-MS, RI
32		Lavandulol acetate	C_12_H_20_O_2_	-	1295	1273	1612	12.68	GC-MS, RI, ^13^C-NMR
33		(*E*)-7-hydroxy-3,7-dimethylocta-1,5-dien-ylacetate	C_12_H_20_O_3_	-	1344	-	-	-	GC-MS, RI, ^13^C-NMR
34		Nerol acetate	C_12_H_20_O_2_	1364	1369	1343	1735	1.07	GC-MS, RI
35		Geranyl acetate	C_12_H_20_O_2_	1387	1387	1361	1765	1.75	GC-MS, RI
36		β-Caryophyllen	C_15_H_24_	1419	1416	1418	1635	1.75	GC-MS, RI
37		α-Santalene	C_15_H_24_	1420	1419	-	-	0.83	GC-MS, RI
38		(*E*)-β-Fanesene	C_15_H_24_	1457	1459	1447	1672	0.49	GC-MS, RI, ^13^C-NMR
39		d-Germacrene	C_15_H_24_	1481	1480	1476	-	0.31	GC-MS, RI, ^13^C-NMR
40		γ-Cadinene	C_15_H_24_	1513	1514	1509	-	0.23	GC-MS, RI, ^13^C-NMR
41		11-hydroxy-α-santal-9-ene	C_15_H_24_O	-	1519	-	-	-	GC-MS, RI, ^13^C-NMR
42		2-Methyl-1-(4-methylphenyl)-3-buten-1-ol	C_12_H_16_O	-	1519	-	-	0.52	GC-MS,RI
43		β-Caryophyllene oxide	C_15_H_24_O	1581	1585	1571	2042	3.65	GC-MS, RI, ^13^C-NMR
44		Cedrelanol	C_15_H_26_O	1640	1643	1628	2206	0.89	GC-MS,RI
45		(3*S*,6*R*,9*R*)-2-(hydroxymethyl)-5,5,9-trimethyltricyclo [7.2.0.0(3,6)]undecan-2-ol	C_15_H_26_O_2_	-	1811	-	-	-	GC-MS, RI, ^13^C-NMR
46		(6*R*,10*R*)-6,10,14-Trimethyl-2-pentadecanone	C_18_H_36_O	-	1846	-	-	-	GC-MS, RI, ^13^C-NMR
47		S_8_		-	2031	-	-	-	GC-MS
Component group									
Monoterpene hydrocarbons								8.03	
Sesquiterpene hydrocarbons								3.61	
Oxygenated monoterpenes								31.53	
Oxygenated sesquiterpenes								4.54	
	Esters							43.23	
	Others							1.14	
	Total identified							92.08	

Bold type indicates major component. The blue type represents the newly identified compound. ^a^ Literature retention indices (HP-5 column) according to NIST 14 (National Institute of Standards and Technology, USA; 14th edition) library database (https://webbook.nist.gov); ^b^ Experiment retention indices (HP-5 ms column); ^c^ Relative abundance calculated on HP-5 ms capillary column.

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
