# Peer review of "Study on Lavender Essential Oil Chemical Compositions by GC-MS and Improved pGC"

_molecules, 2020, doi:10.3390/molecules25143166_

Round 1
Reviewer 1 Report
In my opinion, the work is very interesting. New type of research techniques are used in this work. Chromatographic analyzes are carefully designed and performed.From chromatographic point of view this is very good work. I propose accept in present form.
Author Response
We thank the reviewer for the very helpful comments. Comment 1: In my opinion, the work is very interesting. New type of research techniques are used in this work. Chromatographic analyzes are carefully designed and performed. From chromatographic point of view this is very good work. I propose accept in present form. Response to Comment 1: I would like to express my sincere thanks to reviewer 1 for his recognition of the article. Thank you very much. We thank the reviewer and remain at your disposal for any further question. Your sincerely, Dr. Maitinuer Maiwulanjiang

Reviewer 2 Report
The present study is aimed for the characterization of the essential oil composition from lavender. The study has scientific potential, just improvement is necessary before acceptance.
Authors claim, that newly detected sulphur (S8) compound may be responsible for insecticidal activity of lavender essential oil. In my opinion, such information should not be written neither in the abstract nor in conclusions, as (1) the authors did not performed evaluation of insecticidal activity of analysed essential oil, and (2) amount of this compound is unknown, i.e. not provided (just I did not understand why?).
Other question: what about reproducibility and repeatability of your results? I did not find any information, how many times did your repeat your essential oils extraction, fractionation and GC analyses?
Other comments, please see, in the attached file.

Author Response
Reviewer 2
We thank the reviewer for the very helpful comments. We have studied reviewers’ comment very carefully and revised the manuscript according to the recommendations. Follows are our answers to the referee’s comments.
Comment 1: Authors claim, that newly detected sulphur (S8) compound may be responsible for insecticidal activity of lavender essential oil. In my opinion, such information should not be written neither in the abstract nor in conclusions, as (1) the authors did not performed evaluation of insecticidal activity of analysed essential oil, and (2) amount of this compound is unknown, i.e. not provided (just I did not understand why?).
Response to Comment 1: According to comment 1 of reviewer 2, we had deleted the corresponding contents in the abstract (line 21-23) and conclusions (line 209-212).
Comment 2: what about reproducibility and repeatability of your results? I did not find any information, how many times did your repeat your essential oils extraction, fractionation and GC analyses?
Response to Comment 2: Because our experimental design used GC-MS with three different capillary columns to analyze lavender essential oils, so one combination only analyzed essential oils once. The analysis results of different GC-MS with different columns were different, so the relative content of lavender essential oils component was processed by the analysis of the GC-QTOF-MS with HP-5 ms capillary column. The lavender essence oil was extracted three times with 1200g of material each time. The yields were 12.9852g, 13.9137g and 11.5129g, respectively. Corresponding modifications had also been made in articles (line 132-133). According to your suggestion, we added the detailed separation process of lavender essential oil (line 170-192).
We thank the reviewer and remain at your disposal for any further question.
Your sincerely,
Dr. Maitinuer Maiwulanjiang

Reviewer 3 Report
The manuscript lacks in clear explanation of the material and method. The interest is questionable.
In supplementary figures, the relative abundance of some peaks differ between the columns. Example peak 27 is the highest on HP5 but very small on HP1, why?
How were you able to calculate a peak area when coelution exits
The semi-quantification using peak area in these conditions is not valid
Line 86-88: there is no sulfur compound in the list in Table 1.
Table 1:
The lists of identified compounds has to be compared to the compounds already identified in the literature in lavendula, what is new? What is common?
The identified compounds have to be validated by injection of standards in the same conditions.
beta-famesene should be beta-farnesene
The identification of galantamine is questionable and has to be validated by comparison with a pure standard compound injected in the same conditions.
Line 170: What is the interest to calculate binding energy with AChE?
The conclusion indicates that compounds 39, 40 45 have huge development prospects according to their minimum binding energy, but they are present in only very small amount in the oil, which is not suitable for their pharmaceutical use.
There is also no strong evidence of the added value of pGC.
Author Response
Reviewer 3
We thank the reviewer for the very helpful comments. We have studied reviewers’ comment very carefully and revised the manuscript according to the recommendations. Follows are our answers to the referee’s comments.
Comment 1: The manuscript lacks in clear explanation of the material and method. The interest is questionable.
Response to Comment 1: The source of the material is indicated in lines 122-125 of the revised manuscript, and the extraction of essential oils is supplemented in lines 132-133. According to your suggestion, we added the detailed separation process of lavender essential oil to the revised manuscript (line 170-192).
Comment 2: In supplementary figures, the relative abundance of some peaks differ between the columns. Example peak 27 is the highest on HP-5 but very small on HP1, why?
Response to Comment 2: The relative content data of lavender essential oil in Table 1 were obtained according to the analysis of essential oil by GC-QTOF-MS with HP-5 ms capillary column. Later, in order to increase the qualitative accuracy, essential oil was analyzed by GC-MS with HP-1 and HP-INNOWax capillary column. The above results may be caused by the fact that the samples were not in the same batch, but the qualitative results were not affected.
Comment 3: How were you able to calculate a peak area when coelution exits.
Response to Comment 3: If two or many different components coelution exited, according to the retention index and the matching degree of the quality map, one of the components of the common wash peak is determined, and the peak area of the total wash peak is calculated by the peak area of the component. So we use different polar capillary to analyze lavender essential oil, and to improve the accuracy of each peak.
Comment 4: The semi-quantification using peak area in these conditions is not valid
Response to Comment 4: I'm sorry that our research group does not have the standard, so there is no condition for quantitative analysis.
Comment 5: Line 86-88: there is no sulfur compound in the list in Table 1.
Response to Comment 5: Because the amount of sulfur compounds in lavender essential oil is too low, the sulfur compounds obtained from the 30g essential oil are only enough for analysis, so the structure of the sulfur compounds is not determined. It is only known that there are the compounds in the essential oil through GC-MS. This sulfur compound was shown in Table 1, compound 47.
Comment 6: The lists of identified compounds has to be compared to the compounds already identified in the literature in lavendula, what is new? What is common?
Response to Comment 6: In response to your suggestion, we have made the corresponding changes in Table 1. Bold type indicates major component and the blue type represents the newly identified compound (line 97-98). And a new reference [35] was added (line 315-319).
Comment 7: The identified compounds have to be validated by injection of standards in the same conditions.
Response to Comment 7: we’re sorry that we have no condition to do the experiment because the research group has no standard.
Comment 8: beta-famesene should be beta-farnesene
Response to Comment 8: Thank you for your correction, we had corrected this mistake.
Comment 9: The identification of galantamine is questionable and has to be validated by comparison with a pure standard compound injected in the same conditions.
Response to Comment 9: We're sorry for causing you trouble. Galantamine is put in Table 1 as a positive control for molecular docking, not an essential oil component. Because the content of molecular docking was deleted according to the suggestion of reviewer 4, the relevant content of galantamine has been deleted.
Comment 10: Line 170: What is the interest to calculate binding energy with AChE?
Response to Comment 10: Because acetylcholinesterase inhibitor could be related with Alzheimer disease, the essential oil component is a small molecule and possesses low polarity, can effectively pass through the blood-brain barrier. And lavender essential oil has good nerve activity, so we wanted to screen active compounds through molecular docking. However, we had adopted the suggestion of reviewer 4 to delete the content of molecular docking.
Comment 11: The conclusion indicates that compounds 39, 40 45 have huge development prospects according to their minimum binding energy, but they are present in only very small amount in the oil, which is not suitable for their pharmaceutical use.
Response to Comment 11: If compounds 39, 40 and 45 are proved to have a good inhibitory effect on acetylcholinesterase, the problem of low content of compounds could be solved through chemical synthesis. However, according to the other reviewer’s comment, docking results were removed from the manuscript.
Comment 12: There is also no strong evidence of the added value of pGC.
Response to Comment 12: Compared with the original pGC, the improved GC has greatly reduced the cost of separation and saved a large amount of scientific research funds for the research group.
We thank the reviewer and remain at your disposal for any further question.
Your sincerely,
Dr. Maitinuer Maiwulanjiang
Reviewer 4 Report
The author described that study on lavender essential oil chemical 2 compositions by GC-MS, improved pGC and 3 autodock.
I think that the author explained the introduction of background and the methodology of analysis to detect the GC-MS results.
And also, the author revealed that there are adavantages to use the separation of lavender essential oils by pGC in the early stage.
However, the later section, 2.3. Molecular docking results; That's quite unnecessary to add.
The modeling studies are quite rough, unclear certainty and application of the modeling.
I think this manuscript should be focused on the studies of GC-MS and improved pGC, and reconstructed to clear story.
After the correction of clear story, this manuscript might be suitable for the publication of "Molecules".
Author Response
Reviewer 4
We thank the reviewer for the very helpful comments. We have studied reviewers’ comment very carefully and revised the manuscript according to the recommendations. Follows are our answers to the referee’s comments.
The author described that study on lavender essential oil chemical 2 compositions by GC-MS, improved pGC and 3 autodock. I think that the author explained the introduction of background and the methodology of analysis to detect the GC-MS results. And also, the author revealed that there are adavantages to use the separation of lavender essential oils by pGC in the early stage.
Comment 1: However, the later section, 2.3. Molecular docking results; That's quite unnecessary to add. The modeling studies are quite rough, unclear certainty and application of the modeling.
Response to Comment 1: After careful consideration, we have decided to adopt your suggestion and have deleted the content of molecular docking according to your suggestion.
Comment 2: I think this manuscript should be focused on the studies of GC-MS and improved pGC, and reconstructed to clear story. After the correction of clear story, this manuscript might be suitable for the publication of "Molecules".
Response to Comment 2: According to your suggestion, the content of the manuscript had been adjusted.
We thank the reviewer and remain at your disposal for any further question.
Your sincerely,
Dr. Maitinuer Maiwulanjiang

Round 2
Reviewer 2 Report
Most of mine comments, mentioned in the attached file, were not corrected. I am attaching the old paper with comments.
There is a new question: what was the match of MS spectra of your compounds with MS in NIST library?
Author Response
Reviewer 2
We thank the reviewer for the very helpful comments. We have studied reviewers’ comment very carefully and revised the manuscript according to the recommendations. Follows are our answers to the referee’s comments.
Comment 1: Most of mine comments, mentioned in the attached file, were not corrected. I am attaching the old paper with comments.
Response to Comment 1: Thank you very much again for your review comments. We revised the manuscript according to your advice point by point and changes were highlighted in track tracing mode. Follows are the list of changes that we made.
- P1, line 4-5, we deleted “and” and added “,”.
- P1-2 and P4, line 41-44 and line 93-96, we removed “In the early stage … greatly reduced costs” (line 41-44) to the Results and Discussion (line 93-96).
- P2, line 55, we revised “V/m” to “V/w”.
- P2, line 57 and 75, we revised “table” to “Table”, “figure” to “Figure”.
- P2, line 62, we revised “98.25” to “95.03”.
- P2-4, line 67-71 and Table 1, we revised three numbers to two numbers of digits in the paper.
- P2, line 86-87, we revised the title of the table 1 “Chemical … oil.” to “Relative … oil”.
- P3, line 10, “-” was deleted and the sum of compound 9 and 10 were showed.
- P4, line 102, we revised “$” to “¥”.
- P4, line 113, we added “which was cut into about 2cm”.
- P4, line 114, we added “self-assembling”.
- P4, line 115, we added “and the ratio of material to liquid was 1:10”.
- P5, line 118, 126-127, 135-136, 152-176, we revised “0C” to “℃”.
- P5, line 122, 133-134, 145, we revised “mm” to “μ”.
- P5, line 122, 133-134, 145, we revised “compound 19, 29, 33, 41, 45 and 46” to their full name such as “2,2,6-trimethyl … pentadecanone”.
Comment 2: what was the match of MS spectra of your compounds with MS in NIST library?
Response to Comment 2: The match of MS spectra of the compounds with MS in NIST library is above 80%.
We thank the reviewer and remain at your disposal for any further question.
Your sincerely,
Dr. Maitinuer Maiwulanjiang
Xinjiang Key Laboratory of Plant Resources and Natural Products Chemistry,
Xinjiang Technical Institute of Physics and Chemistry,
Chinese Academy of Sciences,
Beijing South Road 40-1
Urumqi, Xinjiang, 830011, China
Phone: (+86) 0991-6631740
Fax: (+86) 0991-3838957
Email: mavlanjan@ms.xjb.ac.cn

Reviewer 4 Report
As the authors carefully revised the manuscript, I recommend that this manuscript can be suitable for the publication of "Molecules".
Author Response
Reviewer 4
We thank the reviewer for the very helpful comments.
Comment: As the authors carefully revised the manuscript, I recommend that this manuscript can be suitable for the publication of "Molecules".
Response to Comment: I would like to express my sincere thanks to reviewer 4 for his recognition of the article. Thank you very much.
We thank the reviewer and remain at your disposal for any further question.
Your sincerely,
Dr. Maitinuer Maiwulanjiang
Xinjiang Key Laboratory of Plant Resources and Natural Products Chemistry,
Xinjiang Technical Institute of Physics and Chemistry,
Chinese Academy of Sciences,
Beijing South Road 40-1
Urumqi, Xinjiang, 830011, China
Phone: (+86) 0991-6631740
Fax: (+86) 0991-3838957
Email: mavlanjan@ms.xjb.ac.cn
